# A Unified View of Piecewise Linear Neural Network Verification

**Rudy Bunel**
University of Oxford
rudy@robots.ox.ac.uk

**Ilker Turkaslan**
University of Oxford
ilker.turkaslan@lmh.ox.ac.uk

**Philip H.S. Torr**
University of Oxford
philip.torr@eng.ox.ac.uk

**Pushmeet Kohli**
Deepmind
pushmeet@google.com

**M. Pawan Kumar**
University of Oxford
Alan Turing Institute
pawan@robots.ox.ac.uk

## Abstract

The success of Deep Learning and its potential use in many safety-critical applications has motivated research on formal verification of Neural Network (NN) models. Despite the reputation of learned NN models to behave as black boxes and the theoretical hardness of proving their properties, researchers have been successful in verifying some classes of models by exploiting their piecewise linear structure and taking insights from formal methods such as Satisifiability Modulo Theory. These methods are however still far from scaling to realistic neural networks. To facilitate progress on this crucial area, we make two key contributions. First, we present a unified framework that encompasses previous methods. This analysis results in the identification of new methods that combine the strengths of multiple existing approaches, accomplishing a speedup of two orders of magnitude compared to the previous state of the art. Second, we propose a new data set of benchmarks which includes a collection of previously released testcases. We use the benchmark to provide the first experimental comparison of existing algorithms and identify the factors impacting the hardness of verification problems.

## 1 Introduction

Despite their success in a wide variety of applications, Deep Neural Networks have seen limited adoption in safety-critical settings. The main explanation for this lies in their reputation for being black-boxes whose behaviour can not be predicted. Current approaches to evaluate trained models mostly rely on testing using held-out data sets. However, as Edsger W. Dijkstra said [3], "testing shows the presence, not the absence of bugs". If deep learning models are to be deployed to applications such as autonomous driving cars, we need to be able to verify safety-critical behaviours.

To this end, some researchers have tried to use formal methods. To the best of our knowledge, Zakrzewski [20] was the first to propose a method to verify simple, one hidden layer neural networks. However, only recently were researchers able to work with non-trivial models by taking advantage of the structure of ReLU-based networks [4, 11]. Even then, these works are not scalable to the large networks encountered in most real world problems.

This paper advances the field of NN verification by making the following key contributions:

1. We reframe state of the art verification methods as special cases of Branch-and-Bound optimization, which provides us with a **unified framework** to compare them.

2. We gather a data set of test cases based on the existing literature and extend it with new benchmarks. We **provide the first experimental comparison** of verification methods.

3. Based on this framework, we identify algorithmic improvements in the verification process, specifically in the way bounds are computed, the type of branching that are considered, as

well as the strategies guiding the branching. Compared to the previous state of the art, these improvements lead to **speed-up of almost two orders of magnitudes**.

Section 2 and 3 give the specification of the problem and formalise the verification process. Section 4 presents our unified framework, showing that previous methods are special cases and highlighting potential improvements. Section 5 presents our experimental setup and Section 6 analyses the results.

## 2 Problem specification

We now specify the problem of formal verification of neural networks. Given a network that implements a function $\hat{\mathbf{x}}_{\mathbf{n}} = f(\mathbf{x_0})$, a bounded input domain $\mathcal{C}$ and a property $P$, we want to prove

$$\mathbf{x_0} \in \mathcal{C}, \quad \hat{\mathbf{x}}_{\mathbf{n}} = f(\mathbf{x_0}) \implies P(\hat{\mathbf{x}}_{\mathbf{n}}). \tag{1}$$

For example, the property of robustness to adversarial examples in $\mathcal{L}_\infty$ norm around a training sample $\mathbf{a}$ with label $y_a$ would be encoded by using $\mathcal{C} \triangleq \{\mathbf{x_0} | \|\mathbf{x_0} - \mathbf{a}\|_\infty \leq \epsilon\}$ and $P(\hat{\mathbf{x}}_{\mathbf{n}}) = \{\forall y \quad \hat{x}_{n[y_a]} > \hat{x}_{n[y]}\}$.

In this paper, we are going to focus on Piecewise-Linear Neural Networks (PL-NN), that is, networks for which we can decompose $\mathcal{C}$ into a set of polyhedra $\mathcal{C}_i$ such that $\mathcal{C} = \cup_i \mathcal{C}_i$, and the restriction of $f$ to $\mathcal{C}_i$ is a linear function for each $i$. While this prevents us from including networks that use activation functions such as sigmoid or tanh, PL-NNs allow the use of linear transformations such as fully-connected or convolutional layers, pooling units such as MaxPooling and activation functions such as ReLUs. In other words, PL-NNs represent the majority of networks used in practice. Operations such as Batch-Normalization or Dropout also preserve piecewise linearity at test-time.

The properties that we are going to consider are Boolean formulas over linear inequalities. In our robustness to adversarial example above, the property is a conjunction of linear inequalities, each of which constrains the output of the original label to be greater than the output of another label.

The scope of this paper does not include approaches relying on additional assumptions such as twice differentiability of the network [8, 20], limitation of the activation to binary values [4, 15] or restriction to a single linear domain [2]. Since they do not provide formal guarantees, we also don't include approximate approaches relying on a limited set of perturbation [10] or on over-approximation methods that potentially lead to undecidable properties [16, 19].

## 3 Verification Formalism

### 3.1 Verification as a Satisfiability problem

The methods we involve in our comparison all leverage the piecewise-linear structure of PL-NN to make the problem more tractable. They all follow the same general principle: given a property to prove, they attempt to discover a counterexample that would make the property false. This is accomplished by defining a set of variables corresponding to the inputs, hidden units and output of the network, and the set of constraints that a counterexample would satisfy.

To help design a unified framework, we reduce all instances of verification problems to a canonical representation. Specifically, the whole satisfiability problem will be transformed into a global optimization problem where the decision will be obtained by checking the sign of the minimum.

If the property to verify is a simple inequality $P(\hat{\mathbf{x}}_{\mathbf{n}}) \triangleq \mathbf{c}^T \hat{\mathbf{x}}_{\mathbf{n}} > b$, it is sufficient to add to the network a final fully connected layer with one output, with weight of $\mathbf{c}$ and a bias of $-b$. If the global minimum of this network is positive, it indicates that for all $\hat{\mathbf{x}}_{\mathbf{n}}$ the original network can output, we have $\mathbf{c}^T \hat{\mathbf{x}}_{\mathbf{n}} - b > 0 \implies \mathbf{c}^T \hat{\mathbf{x}}_{\mathbf{n}} > b$, and as a consequence the property is True. On the other hand, if the global minimum is negative, then the minimizer provides a counter-example. The supplementary material shows that `OR` and `AND` clauses in the property can similarly be expressed as additional layers, using MaxPooling units.

We can formulate any Boolean formula over linear inequalities on the output of the network as a sequence of additional linear and max-pooling layers. The verification problem will be reduced to the problem of finding whether the scalar output of the potentially modified network can reach a negative value. Assuming the network only contains ReLU activations between each layer, the satisfiability problem to find a counterexample can be expressed as:

$$\mathbf{l_0} \leq \mathbf{x_0} \leq \mathbf{u_0} \tag{2a}$$

$$\hat{\mathbf{x}}_{\mathbf{i+1}} = W_{i+1}\mathbf{x_i} + \mathbf{b_{i+1}} \qquad \forall i \in \{0, \ n-1\} \tag{2c}$$

$$\hat{x}_n \leq 0 \tag{2b}$$

$$\mathbf{x_i} = \max(\hat{\mathbf{x}}_{\mathbf{i}}, 0) \qquad \forall i \in \{1, \ n-1\}. \tag{2d}$$

Eq. (2a) represents the constraints on the input and Eq. (2b) on the neural network output. Eq. (2c) encodes the linear layers of the network and Eq. (2d) the ReLU activation functions. If an assignment to all the values can be found, this represents a counterexample. If this problem is unsatisfiable, no counterexample can exist, implying that the property is True. We emphasise that we are required to prove that no counter-examples can exist, and not simply that none could be found.

While for clarity of explanation, we have limited ourselves to the specific case where only ReLU activation functions are used, this is not restrictive. The supplementary material contains a section detailing how each method specifically handles MaxPooling units, as well as how to convert any MaxPooling operation into a combination of linear layers and ReLU activation functions.

The problem described in (2) is still a hard problem. The addition of the ReLU non-linearities (2d) transforms a problem that would have been solvable by simple Linear Programming into an NP-hard problem [11]. Converting a verification problem into this canonical representation does not make its resolution simpler but it does provide a formalism advantage. Specifically, it allows us to prove complex properties, containing several OR clauses, with a single procedure rather than having to decompose the desired property into separate queries as was done in previous work [11].

Operationally, a valid strategy is to impose the constraints (2a) to (2d) and minimise the value of $\hat{x}_n$. Finding the exact global minimum is not necessary for verification. However, it provides a measure of satisfiability or unsatisfiability. If the value of the global minimum is positive, it will correspond to the margin by which the property is satisfied.

## 3.2 Mixed Integer Programming formulation

A possible way to eliminate the non-linearities is to encode them with the help of binary variables, transforming the PL-NN verification problem (2) into a Mixed Integer Linear Program (MIP). This can be done with the use of "big-M" encoding. The following encoding is from Tjeng & Tedrake [18]. Assuming we have access to lower and upper bounds on the values that can be taken by the coordinates of $\hat{\mathbf{x}}_i$, which we denote $\mathbf{l}_i$ and $\mathbf{u}_i$, we can replace the non-linearities:

$$x_i = \max(\hat{\mathbf{x}}_i, 0) \quad \Rightarrow \quad \boldsymbol{\delta_i} \in \{0,1\}^{h_i}, \quad \mathbf{x_i} \geq 0, \qquad \mathbf{x_i} \leq \mathbf{u_i} \cdot \boldsymbol{\delta_i} \tag{3a}$$

$$\mathbf{x_i} \geq \hat{\mathbf{x}}_i, \qquad \mathbf{x_i} \leq \hat{\mathbf{x}}_i - \mathbf{l_i} \cdot (1 - \boldsymbol{\delta_i}) \tag{3b}$$

It is easy to verify that $\delta_{i[j]} = 0 \Leftrightarrow x_{i[j]} = 0$ (replacing $\delta_{i[j]}$ in Eq. (3a)) and $\delta_{i[j]} = 1 \Leftrightarrow x_{i[j]} = \hat{x}_{i[j]}$ (replacing $\delta_{i[j]}$ in Eq. (3b)).

By taking advantage of the feed-forward structure of the neural network, lower and upper bounds $\mathbf{l_i}$ and $\mathbf{u_i}$ can be obtained by applying interval arithmetic [9] to propagate the bounds on the inputs, one layer at a time.

Thanks to this specific feed-forward structure of the problem, the generic, non-linear, non-convex problem has been rewritten into an MIP. Optimization of MIP is well studied and highly efficient off-the-shelf solvers exist. As solving them is NP-hard, performance is going to be dependent on the quality of both the solver used and the encoding. We now ask the following question: how much efficiency can be gained by using a bespoke solver rather than a generic one? In order to answer this, we present specialised solvers for the PLNN verification task.

# 4 Branch-and-Bound for Verification

As described in Section 3.1, the verification problem can be rephrased as a global optimization problem. Algorithms such as Stochastic Gradient Descent are not appropriate as they have no way of guaranteeing whether or not a minima is global. In this section, we present an approach to estimate the global minimum, based on the Branch and Bound paradigm and show that several published methods, introduced as examples of Satisfiability Modulo Theories, fit this framework.

Algorithm 1 describes its generic form. The input domain is repeatedly split into sub-domains (line 7), over which lower and upper bounds of the minimum are computed (lines 9-10). The best upper-bound found so far serves as a candidate for the global minimum. Any domain whose lower bound is greater than the current global upper bound can be pruned away as it cannot contain the global minimum (line 13, lines 15-17). By iteratively splitting the domains, it is possible to compute tighter lower bounds. We keep track of the global lower bound on the minimum by taking the minimum over the lower bounds of all sub-domains (line 19). When the global upper bound and the global lower bound differ by less than a small scalar $\epsilon$ (line 5), we consider that we have converged.

Algorithm 1 shows how to optimise and obtain the global minimum. If all that we are interested in is the satisfiability problem, the procedure can be simplified by initialising the global upper bound with

0 (in line 2). Any subdomain with a lower bound greater than 0 (and therefore not eligible to contain a counterexample) will be pruned out (by line 15). The computation of the lower bound can therefore be replaced by the feasibility problem (or its relaxation) imposing the constraint that the output is below zero without changing the algorithm. If it is feasible, there might still be a counterexample and further branching is necessary. If it is infeasible, the subdomain can be pruned out. In addition, if any upper bound improving on 0 is found on a subdomain (line 11), it is possible to stop the algorithm as this already indicates the presence of a counterexample.

---

**Algorithm 1** Branch and Bound

```
 1: function BAB(net, domain, ε)
 2:    global_ub ← inf
 3:    global_lb ← − inf
 4:    doms ← [(global_lb, domain)]
 5:    while global_ub − global_lb > ε do
 6:       (_ , dom) ← pick_out(doms)
 7:       [subdom_1, . . . , subdom_s] ← split(dom)
 8:       for i = 1 . . . s do
 9:          dom_ub ← compute_UB(net, subdom_i)
10:          dom_lb ← compute_LB(net, subdom_i)
11:          if dom_ub < global_ub then
12:             global_ub ← dom_ub
13:             prune_domains(doms, global_ub)
14:          end if
15:          if dom_lb < global_ub then
16:             domains.append((dom_lb, subdom_i))
17:          end if
18:       end for
19:       global_lb ← min{lb | (lb, dom) ∈ doms}
20:    end while
21:    return global_ub
22: end function
```

The description of the verification problem as optimization and the pseudo-code of Algorithm 1 are generic and would apply to verification problems beyond the specific case of PL-NN. To obtain a practical algorithm, it is necessary to specify several elements.

**A search strategy**, defined by the `pick_out` function, which chooses the next domain to branch on. Several heuristics are possible, for example those based on the results of previous bound computations. For satisfiable problems or optimization problems, this allows to discover good upper bounds, enabling early pruning.

**A branching rule**, defined by the `split` function, which takes a domain dom and return a partition in subdomain such that $\bigcup_i$ subdom_i = dom and that (subdom_i ∩ subdom_j) = ∅, ∀i ≠ j. This will define the "shape" of the domains, which impacts the hardness of computing bounds. In addition, choosing the right partition can greatly impact the quality of the resulting bounds.

**Bounding methods**, defined by the `compute_{UB, LB}` functions. These procedures estimate respectively upper bounds and lower bounds over the minimum output that the network `net` can reach over a given input domain. We want the lower bound to be as high as possible, so that this whole domain can be pruned easily. This is usually done by introducing convex relaxations of the problem and minimising them. On the other hand, the computed upper bound should be as small as possible, so as to allow pruning out other regions of the space or discovering counterexamples. As any feasible point corresponds to an upper bound on the minimum, heuristic methods are sufficient.

We now demonstrate how some published work in the literature can be understood as special case of the branch-and-bound framework for verification.

### 4.1   Reluplex

Katz et al. [11] present a procedure named Reluplex to verify properties of Neural Network containing linear functions and ReLU activation unit, functioning as an SMT solver using the splitting-on-demand framework [1]. The principle of Reluplex is to always maintain an assignment to all of the variables, even if some of the constraints are violated.

Starting from an initial assignment, it attempts to fix some violated constraints at each step. It prioritises fixing linear constraints ((2a), (2c), (2b) and some relaxation of (2d)) using a simplex algorithm, even if it leads to violated ReLU constraints. If no solution to this relaxed problem containing only linear constraints exists, the counterexample search is unsatisfiable. Otherwise, either all ReLU are respected, which generates a counterexample, or Reluplex attempts to fix one of the violated ReLU; potentially leading to newly violated linear constraints. This process is not guaranteed to converge, so to make progress, non-linearities that get fixed too often are split into two cases. Two new problems are generated, each corresponding to one of the phases of the ReLU. In the worst setting, the problem will be split completely over all possible combinations of activation patterns, at which point the sub-problems will all be simple LPs.

This algorithm can be mapped to the special case of branch-and-bound for satisfiability. The **search strategy** is handled by the SMT core and to the best of our knowledge does not prioritise any domain. The **branching rule** is implemented by the ReLU-splitting procedure: when neither the upper bound

search, nor the detection of infeasibility are successful, one non-linear constraint over the $j$-th neuron of the $i$-th layer $x_{i[j]} = \max\left(\hat{x}_{i[j]}, 0\right)$ is split out into two subdomains: $\{x_{i[j]} = 0, \hat{x}_{i[j]} \leq 0\}$ and $\{x_{i[j]} = \hat{x}_{i[j]}, \hat{x}_{i[j]} \geq 0\}$. This defines the type of subdomains produced. The prioritisation of ReLUs that have been frequently fixed is a heuristic to decide between possible partitions.

As Reluplex only deal with satisfiability, the analogue of the lower bound computation is an over-approximation of the satisfiability problem. The **bounding method** used is a convex relaxation, obtained by dropping some of the constraints. The following relaxation is applied to ReLU units for which the sign of the input is unknown ($l_{i[j]} \leq 0$ and $u_{i[j]} \geq 0$).

$$\mathbf{x_i} = \max\left(\hat{\mathbf{x}}_\mathbf{i}, 0\right) \quad \Rightarrow \quad \mathbf{x_i} \geq \hat{\mathbf{x}}_\mathbf{i} \quad (4a) \qquad \mathbf{x_i} \geq 0 \quad (4b) \qquad \mathbf{x_i} \leq \mathbf{u_i}. \quad (4c)$$

If this relaxation is unsatisfiable, this indicates that the subdomain cannot contain any counterexample and can be pruned out. The search for an assignment satisfying all the ReLU constraints by iteratively attempting to correct the violated ReLUs is a heuristic that is equivalent to the search for an upper bound lower than 0: success implies the end of the procedure but no guarantees can be given.

## 4.2 Planet

Ehlers [6] also proposed an approach based on SMT. Unlike Reluplex, the proposed tool, named Planet, operates by explicitly attempting to find an assignment to the phase of the non-linearities. Reusing the notation of Section 3.2, it assigns a value of 0 or 1 to each $\delta_{i[j]}$ variable, verifying at each step the feasibility of the partial assignment so as to prune infeasible partial assignment early.

As in Reluplex, the **search strategy** is not explicitly encoded and simply enumerates all the domains that have not yet been pruned. The **branching rule** is the same as for Reluplex, as fixing the decision variable $\delta_{i[j]} = 0$ is equivalent to choosing $\{x_{i[j]} = 0, \hat{x}_{i[j]} \leq 0\}$ and fixing $\delta_{i[j]} = 1$ is equivalent to $\{x_{i[j]} = \hat{x}_{i[j]}, \hat{x}_{i[j]} \geq 0\}$. Note however that Planet does not include any heuristic to prioritise which decision variables should be split over.

Planet does not include a mechanism for early termination based on a heuristic search of a feasible point. For satisfiable problems, only when a full complete assignment is identified is a solution returned. In order to detect incoherent assignments, Ehlers [6] introduces a global linear approximation to a neural network, which is used as a **bounding method** to over-approximate the set of values that each hidden unit can take. In addition to the existing linear constraints ((2a), (2c) and (2b)), the non-linear constraints are approximated by sets of linear constraints representing the non-linearities' convex hull. Specifically, ReLUs with input of unknown sign are replaced by the set of equations:

$$\mathbf{x_i} = \max\left(\hat{\mathbf{x}}_\mathbf{i}, 0\right) \Rightarrow \mathbf{x_i} \geq \hat{\mathbf{x}}_\mathbf{i} \quad (5a) \qquad \mathbf{x_i} \geq 0 \quad (5b) \qquad x_{i[j]} \leq u_{i[j]} \frac{\hat{x}_{i[j]} - l_{i[j]}}{u_{i[j]} - l_{i[j]}} \quad (5c)$$

where $x_{i[j]}$ corresponds to the value of the $j$-th coordinate of $\mathbf{x_i}$. An illustration of the feasible domain is provided in the supplementary material.

Compared with the relaxation of Reluplex (4), the Planet relaxation is tighter. Specifically, Eq. (4a) and (4b) are identical to Eq. (5a) and (5b) but Eq. (5c) implies Eq. (4c). Indeed, given that $\hat{x}_{i[j]}$ is smaller than $u_{i[j]}$, the fraction multiplying $u_{i[j]}$ is necessarily smaller than 1, implying that this provides a tighter bounds on $x_{i[j]}$.

To use this approximation to compute better bounds than the ones given by simple interval arithmetic, it is possible to leverage the feed-forward structure of the neural networks and obtain bounds one layer at a time. Having included all the constraints up until the $i$-th layer, it is possible to optimize over the resulting linear program and obtain bounds for all the units of the $i$-th layer, which in turn will allow us to create the constraints (5) for the next layer.

In addition to the pruning obtained by the convex relaxation, both Planet and Reluplex make use of conflict analysis [14] to discover combinations of splits that cannot lead to satisfiable assignments, allowing them to perform further pruning of the domains.

## 4.3 Potential improvements

As can be seen, previous approaches to neural network verification have relied on methodologies developed in three communities: optimization, for the creation of upper and lower bounds; verification, especially SMT; and machine learning, especially the feed-forward nature of neural networks for the creation of relaxations. A natural question that arises is "Can other existing literature from these domains be exploited to further improve neural network verification?" Our unified branch-and-bound formulation makes it easy to answer this question. To illustrate its power, we now provide a non-exhaustive list of suggestions to speed-up verification algorithms.

**Better bounding —**  While the relaxation proposed by Ehlers [6] is tighter than the one used by Reluplex, it can be improved further still. Specifically, after a splitting operation, on a smaller domain, we can refine all the $\mathbf{l_i}$, $\mathbf{u_i}$ bounds, to obtain a tither relaxation. We show the importance of this in the experiments section with the **BaB-relusplit** method that performs splitting on the activation like Planet but updates its approximation completely at each step.

One other possible area of improvement lies in the tightness of the bounds used. Equation (5) is very closely related to the Mixed Integer Formulation of Equation (3). Indeed, it corresponds to level 0 of the Sherali-Adams hierarchy of relaxations [17]. The proof for this statement can be found in the supplementary material. Stronger relaxations could be obtained by exploring higher levels of the hierarchy. This would jointly constrain groups of ReLUs, rather than linearising them independently.

**Better branching**  The decision to split on the activation of the ReLU non-linearities made by Planet and Reluplex is intuitive as it provides a clear set of decision variables to fix. However, it ignores another natural branching strategy, namely, splitting the input domain. Indeed, it could be argued that since the function encoded by the neural networks are piecewise linear in their input, this could result in the computation of highly useful upper and lower bounds. To demonstrate this, we propose the novel **BaB-input** algorithm: a branch-and-bound method that branches over the input features of the network. Based on a domain with input constrained by Eq. (2a), the `split` function would return two subdomains where bounds would be identical in all dimension except for the dimension with the largest length, denoted $i^\star$. The bounds for each subdomain for dimension $i^\star$ are given by $l_{0[i^\star]} \leq x_{0[i^\star]} \leq \frac{l_{0[i^\star]}+u_{0[i^\star]}}{2}$ and $\frac{l_{0[i^\star]}+u_{0[i^\star]}}{2} \leq x_{0[i^\star]} \leq u_{0[i^\star]}$. Based on these tighter input bounds, tighter bounds at all layers can be re-evaluated.

One of the main advantage of branching over the variables is that all subdomains generated by the BaB algorithm when splitting over the input variables end up only having simple bound constraints over the value that input variable can take. In order to exploit this property to the fullest, we use the highly efficient lower bound computation approach of Kolter & Wong [13]. This approach was initially proposed in the context of robust optimization. However, our unified framework opens the door for its use in verification. Specifically, Kolter & Wong [13] identified an efficient way of computing bounds for the type of problems we encounter, by generating a feasible solution to the dual of the LP generated by the Planet relaxation. While this bound is quite loose compared to the one obtained through actual optimization, they are very fast to evaluate. We propose a smart branching method **BaBSB** to replace the longest edge heuristic of **BaB-input**. For all possible splits, we compute fast bounds for each of the resulting subdomain, and execute the split resulting in the highest lower bound. The intuition is that despite their looseness, the fast bounds will still be useful in identifying the promising splits.

Another advantage of the branch-and-bound approach is that it's not dependent on the networks being piecewise linear. While methods such as **Reluplex**, **Planet** or the MIP encoding depends on the piecewise linearity, any type of networks for which an appropriate bounding function can be found will be verifiable using branch-and-bound. Recent advances on incomplete verification such as the work of Dvijotham et al. [5] can offer such bounds for activations such as sigmoid or hyperbolic tangent.

## 5  Experimental setup

The problem of PL-NN verification has been shown to be NP-complete [11]. Meaningful comparison between approaches therefore needs to be experimental.

### 5.1  Methods

The simplest baseline we refer to is **BlackBox**, a direct encoding of Eq. (2) into the Gurobi solver, which will perform its own relaxation, without taking advantage of the problem's structure.

For the SMT based methods, **Reluplex** and **Planet**, we use the publicly available versions [7, 12]. Both tools are implemented in C++ and relies on the GLPK library to solve their relaxation. We wrote some software to convert in both directions between the input format of both solvers.

We also evaluate the potential of using MIP solvers, based on the formulation of Eq. (3). Due to the lack of availability of open-sourced methods at the time of our experiments, we reimplemented the approach in Python, using the Gurobi MIP solver. We report results for a variant called **MIPplanet**, which uses bounds derived from Planet's convex relaxation rather than simple interval arithmetic. Both the MIP and **BlackBox** are not treated as simple feasibility problem but are encoded to minimize the

output $\hat{x}_n$ of Equation (2b), with a callback interrupting the optimization as soon as a negative value is found. Additional discussions on encodings of the MIP problem can be found in supplementary materials.

In our benchmark, we include the methods derived from our Branch and Bound analysis. Our implementation follows faithfully Algorithm 1, is implemented in Python and uses Gurobi to solve LPs. The `pick_out` strategy consists in prioritising the domain that currently has the smallest lower bound. Upper bounds are generated by randomly sampling points on the considered domain, and we use the convex approximation of Ehlers [6] to obtain lower bounds. As opposed to the approach taken by Ehlers [6] of building a single approximation of the network, we rebuild the approximation and recompute all bounds for each sub-domain. This is motivated by the observation shown in Figure 1 which demonstrate the significant improvements it brings, especially for deeper networks. For `split`, **BaB-input** performs branching by splitting the input domain in half along its longest edge and **BaBSB** does it by splitting the input domain along the dimension improving the global lower bound the most according to the fast bounds of Kolter & Wong [13]. We also include results for the **BaB-relusplit** variant, where the `split` method is based on the phase of the non-linearities, similarly to **Planet**.

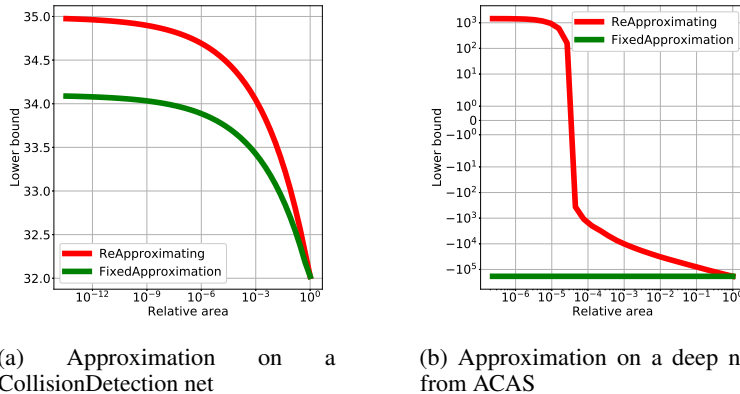

(a)    Approximation    on    a CollisionDetection net

(b) Approximation on a deep net from ACAS

Figure 1: *Quality of the linear approximation, depending on the size of the input domain. We plot the value of the lower bound as a function of the area on which it is computed (higher is better). The domains are centered around the global minimum and repeatedly shrunk. Rebuilding completely the linear approximation at each step allows to create tighter lower-bounds thanks to better $\mathbf{l_i}$ and $\mathbf{u_i}$, as opposed to using the same constraints and only changing the bounds on input variables. This effect is even more significant on deeper networks.*

## 5.2    Evaluation Criteria

For each of the data sets, we compare the different methods using the same protocol. We attempt to verify each property with a timeout of two hours, and a maximum allowed memory usage of 20GB, on a single core of a machine with an i7-5930K CPU. We measure the time taken by the solvers to either prove or disprove the property. If the property is false and the search problem is therefore satisfiable, we expect from the solver to exhibit a counterexample. If the returned input is not a valid counterexample, we don't count the property as successfully proven, even if the property is indeed satisfiable. All code and data necessary to replicate our analysis are released.

## 6    Analysis

We attempt to perform verification on three data sets of properties and report the comparison results. The dimensions of all the problems can be found in the supplementary material.

The **CollisionDetection** data set [6] attempts to predict whether two vehicles with parameterized trajectories are going to collide. 500 properties are extracted from problems arising from a binary search to identify the size of the region around training examples in which the prediction of the network does not change. The network used is relatively shallow but due to the process used to generate the properties, some lie extremely close between the decision boundary between SAT and UNSAT. Results presented in Figure 2 therefore highlight the accuracy of methods.

The **Airborne Collision Avoidance System (ACAS)** data set, as released by Katz et al. [11] is a neural network based advisory system recommending horizontal manoeuvres for an aircraft in order to avoid collisions, based on sensor measurements. Each of the five possible manoeuvres is assigned

a score by the neural network and the action with the minimum score is chosen. The 188 properties to verify are based on some specification describing various scenarios. Due to the deeper network involved, this data set is useful in highlighting the scalability of the various algorithms.

Existing data sets do not allow us to explore the impact on the performance of different methods of various problem/model parameters such as depth, number of hidden units, and input dimensionality. Our new data set, **PCAMNIST**, removes this deficiency, and can prove helpful in analysing future verification approaches as well. It is generated in a way to give control over different architecture parameters. Details of the dataset construction are given in supplementary materials. We present plots in Figure 4 showing the evolution of runtimes depending on each of the architectural parameters of the networks.

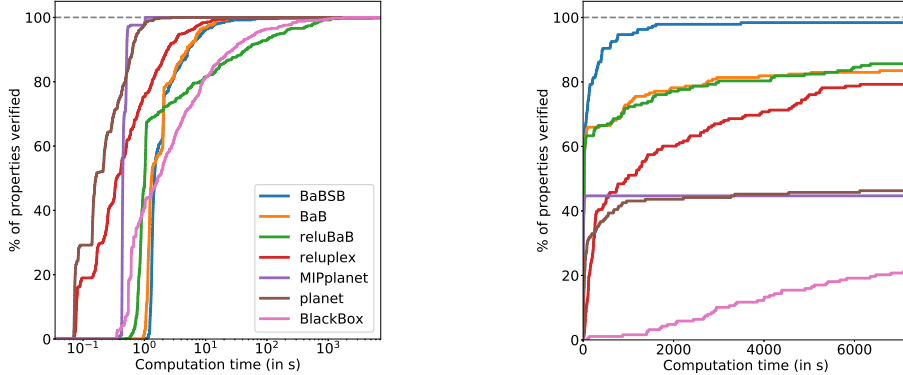

(a) *CollisionDetection Dataset*                    (b) *ACAS Dataset*

Figure 2: *Proportion of properties verifiable for varying time budgets depending on the methods employed. A higher curve means that for the same time budget, more properties will be solvable. All methods solve CollisionDetection quite quickly except **reluBaB**, which is much slower and **BlackBox** who produces several incorrect counterexamples.*

| Method | Average time per Node |
|---|---|
| BaBSB | 1.81s |
| BaB | 2.11s |
| reluBaB | 1.69s |
| reluplex | 0.30s |
| MIPplanet | 0.017s |
| planet | 1.5e-3s |

Table 1: *Average time to explore a node for each method.*

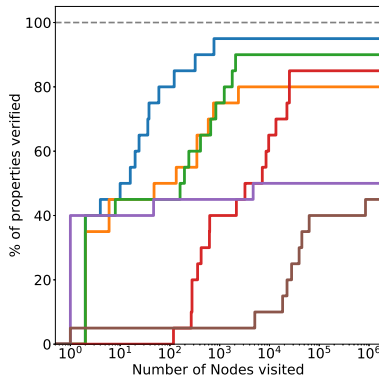

(a) *Properties solved for a given number of nodes to explore (log scale).*

Figure 3: *The trade-off taken by the various methods are different. Figure 3a shows how many subdomains needs to be explored before verifying properties while Table 1 shows the average time cost of exploring each subdomain. Our methods have a higher cost per node but they require significantly less branching, thanks to better bounding. Note also that between **BaBSB** and **BaB**, the smart branching reduces by an order of magnitude the number of nodes to visit.*

**Comparative evaluation of verification approaches —** In Figure 2, on the shallow networks of **CollisionDetection**, most solvers succeed against all properties in about 10s. In particular, the SMT inspired solvers **Planet**, **Reluplex** and the MIP solver are extremely fast.

On the deeper networks of **ACAS**, in Figure 2b, no errors are observed but most methods timeout on the most challenging testcases. The best baseline is **Reluplex**, who reaches 79.26% success rate at the two hour timeout, while our best method, **BaBSB**, already achieves 98.40% with a budget of one hour. To reach Reluplex's success rate, the required runtime is two orders of magnitude smaller.

**Impact of each improvement —** To identify which changes allow our method to have good performance, we perform an ablation study and study the impact of removing some components of our methods. The only difference between **BaBSB** and **BaB** is the smart branching, which represents a significant part of the performance gap.

Branching over the inputs rather than over the activations does not contribute much, as shown by the small difference between **BaB** and **reluBaB**. Note however that we are able to use the fast methods of Kolter & Wong [13] for the smart branching because branching over the inputs makes the bounding problem similar to the one solved in robust optimization. Even if it doesn't improve performance by itself, the new type of split enables the use of smart branching.

The rest of the performance gap can be attributed to using a better bounds: **reluBaB** significantly outperforms **planet** while using the same branching strategy and the same convex relaxations. The improvement comes from the benefits of rebuilding the approximation at each step shown in Figure 1.

Figure 3 presents some additional analysis on a 20-property subset of the ACAS dataset, showing how the methods used impact the need for branching. Smart branching and the use of better lower bounds reduce heavily the number of subdomains to explore.

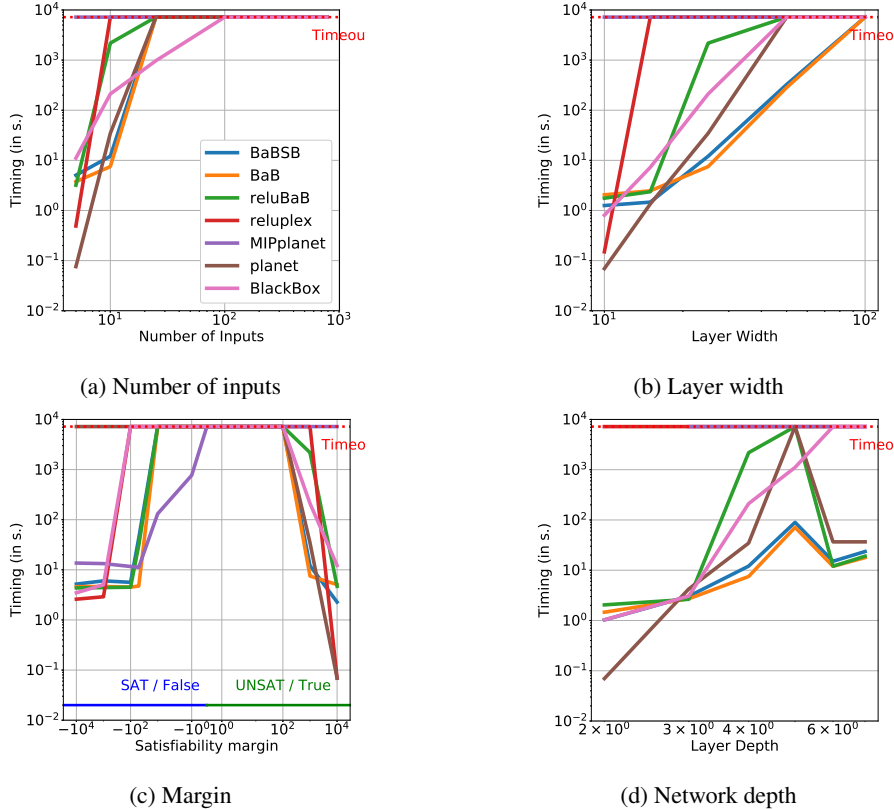

(a) Number of inputs          (b) Layer width

(c) Margin          (d) Network depth

Figure 4: *Impact of the various parameters over the runtimes of the different solvers. The base network has 10 inputs and 4 layers of 25 hidden units, and the property to prove is True with a margin of 1000. Each of the plot correspond to a variation of one of this parameters.*

In the graphs of Figure 4, the trend for all the methods are similar, which seems to indicate that hard properties are intrinsically hard and not just hard for a specific solver. Figure 4a shows an expected trend: the largest the number of inputs, the harder the problem is. Similarly, Figure 4b shows unsurprisingly that wider networks require more time to solve, which can be explained by the fact that they have more non-linearities. The impact of the margin, as shown in Figure 4c is also clear. Properties that are True or False with large satisfiability margin are easy to prove, while properties that have small satisfiability margins are significantly harder.

# 7    Conclusion

The improvement of formal verification of Neural Networks represents an important challenge to be tackled before learned models can be used in safety critical applications. By providing both a unified framework to reason about methods and a set of empirical benchmarks to measure performance with, we hope to contribute to progress in this direction. Our analysis of published algorithms through the lens of Branch and Bound optimization has already resulted in significant improvements in runtime on our benchmarks. Its continued analysis should reveal even more efficient algorithms in the future.

## 8 Acknowldgments

This work was supported by ERC grant ERC-2012-AdG 321162-HELIOS, EPSRC grant Seebibyte EP/M013774/1 and EPSRC/MURI grant EP/N019474/1. We would also like to acknowledge the Royal Academy of Engineering and FiveAI.

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
