[Supplementary Material · nn_verif_supplementary.pdf]

# A Unified View of Piecewise Linear Neural Network Verification Supplementary Materials

**Rudy Bunel**
University of Oxford
rudy@robots.ox.ac.uk

**Ilker Turkaslan**
University of Oxford
ilker.turkaslan@lmh.ox.ac.uk

**Philip H.S. Torr**
University of Oxford
philip.torr@eng.ox.ac.uk

**Pushmeet Kohli**
Deepmind
pushmeet@google.com

**M. Pawan Kumar**
University of Oxford
Alan Turing Institute
pawan@robots.ox.ac.uk

## 1   Canonical Form of the verification problem

If the property is a simple inequality $P(\hat{\mathbf{x}}_\mathbf{n}) \triangleq \mathbf{c}^T\hat{\mathbf{x}}_\mathbf{n} > b$, it is sufficient to add to the network a final fully connected layer with one output, with weight of $\mathbf{c}$ and a bias of $-b$. If the global minimum of this network is positive, it indicates that for all $\hat{\mathbf{x}}_\mathbf{n}$ the original network can output, we have $\mathbf{c}^T\hat{\mathbf{x}}_\mathbf{n} - b > 0 \implies \mathbf{c}^T\hat{\mathbf{x}}_\mathbf{n} > b$, and as a consequence the property is True. On the other hand, if the global minimum is negative, then the minimizer provides a counter-example.

Clauses specified using OR (denoted by $\bigvee$) can be encoded by using a MaxPooling unit. If the property is $P(\hat{\mathbf{x}}_\mathbf{n}) \triangleq \bigvee_i \left[\mathbf{c}_i^T\hat{\mathbf{x}}_\mathbf{n} > b_i\right]$, this is equivalent to $\max_i \left(\mathbf{c}_i^T\hat{\mathbf{x}}_\mathbf{n} - b_i\right) > 0$.

Clauses specified using AND (denoted by $\bigwedge$) can be encoded similarly: the property $P(\hat{x}_n) = \bigwedge_i \left[\mathbf{c}_i^T\hat{\mathbf{x}}_\mathbf{n} > b_i\right]$ is equivalent to $\min_i \left(\mathbf{c}_i^T\hat{\mathbf{x}}_\mathbf{n} - b_i\right) > 0 \iff -\left(\max_i \left(-\mathbf{c}_i^T\hat{\mathbf{x}}_\mathbf{n} + b_i\right)\right) > 0$

## 2   Toy Problem example

We have specified the problem of formal verification of neural networks as follows: given a network that implements a function $\hat{\mathbf{x}}_\mathbf{n} = f(\mathbf{x_0})$, a bounded input domain $\mathcal{C}$ and a property $P$, we want to prove that

$$\mathbf{x_0} \in \mathcal{C}, \quad \hat{\mathbf{x}}_\mathbf{n} = f(\mathbf{x_0}) \implies P(\hat{\mathbf{x}}_\mathbf{n}). \tag{1}$$

A toy-example of the Neural Network verification problem is given in Figure 1. On the domain $\mathcal{C} = [-2; 2] \times [-2; 2]$, we want to prove that the output $y$ of the one hidden-layer network always satisfy the property $P(y) \triangleq [y > -5]$. We will use this as a running example to explain the methods used for comparison in our experiments.

a

[-2, 2]$x_1$

-1

$y$

-1

[-2, 2]$x_2$

-1

b

Prove that $y > -5$

Figure 1: Example Neural Network. We attempt to prove the property that the network output is always greater than -5

## 2.1 Problem formulation

For the network of Figure 1, the variables would be $\{x_1, x_2, a_{\text{in}}, a_{\text{out}}, b_{\text{in}}, b_{\text{out}}, y\}$ and the set of constraints would be:

$$-2 \le x_1 \le 2 \qquad\qquad -2 \le x_2 \le 2 \tag{2a}$$

$$\hat{a} = x_1 + x_2 \qquad\qquad \hat{b} = -x_1 - x_2$$

$$y = -a - b \tag{2b}$$

$$a = \max(\hat{a}, 0) \qquad\qquad b = \max(\hat{b}, 0) \tag{2c}$$

$$y \le -5 \tag{2d}$$

Here, $\hat{a}$ is the input value to hidden unit, while $a$ is the value after the ReLU. Any point satisfying all the above constraints would be a counterexample to the property, as it would imply that it is possible to drive the output to -5 or less.

## 2.2 MIP formulation

In our example, the non-linearities of equation (2c) would be replaced by

$$a \ge 0 \qquad\qquad a \ge \hat{a}$$

$$a \le \hat{a} - l_a(1 - \delta_a) \qquad\qquad a \le u_a \delta_a \tag{3}$$

$$\delta_a \in \{0, 1\}$$

where $l_a$ is a lower bound of the value that $\hat{a}$ can take (such as -4) and $u_a$ is an upper bound (such as 4). The binary variable $\delta_a$ indicates which phase the ReLU is in: if $\delta_a = 0$, the ReLU is blocked and $a_{\text{out}} = 0$, else the ReLU is passing and $a_{\text{out}} = a_{\text{in}}$. The problem remains difficult due to the integrality constraint on $\delta_a$.

## 2.3 Running Reluplex

Table 2 shows the initial steps of a run of the Reluplex algorithm on the example of Figure 1. Starting from an initial assignment, it attempts to fix some violated constraints at each step. It prioritises fixing linear constraints ((2a), (2b) and (2d) in our illustrative example) using a simplex algorithm, even if it leads to violated ReLU constraints (2c). This can be seen in step 1 and 3 of the process.

If no solution to the problem containing only linear constraints exists, this shows that the counterexample search is unsatisfiable. Otherwise, all linear constraints are fixed and Reluplex attempts to fix one violated ReLU at a time, such as in step 2 of Table 2 (fixing the ReLU $b$), potentially leading to newly violated linear constraints. In the case where no violated ReLU exists, this means that a satisfiable assignment has been found and that the search can be interrupted.

This process is not guaranteed to converge, so to guarantee progress, non-linearities getting fixed too often are split into two cases. This generates two new sub-problems, each involving an additional

| Step | $x_1$ | $x_2$ | $\hat{a}$ | $a$ | $\hat{b}$ | $b$ | $y$ |
|---|---|---|---|---|---|---|---|
| | 0 | 0 | 0 | 0 | 0 | 0 | 0 |
| 1 | | | Fix linear constraints | | | | |
| | 0 | 0 | 0 | 1 | 0 | 4 | -5 |
| | 0 | 0 | 0 | 1 | 0 | 4 | -5 |
| 2 | | | Fix a ReLU | | | | |
| | 0 | 0 | 0 | 1 | 4 | 4 | -5 |
| | 0 | 0 | 0 | 1 | 4 | 4 | -5 |
| 3 | | | Fix linear constraints | | | | |
| | -2 | -2 | -4 | 1 | 4 | 4 | -5 |
| | | | $\cdots$ | | | | |

Figure 2: *Evolution of the Reluplex algorithm. Red cells corresponds to value violating Linear constraints, and orange cells corresponds to value violating ReLU constraints. Resolution of violation of linear constraints are prioritised.*

linear constraint instead of the linear one. The first one solves the problem where $\hat{a} \leq 0$ and $a = 0$, the second one where $\hat{a} \geq 0$ and $a = \hat{a}$. In the worst setting, the problem will be split completely over all possible combinations of activation patterns, at which point the sub-problems are simple LPs.

## 3   Planet approximation

The feasible set of the Mixed Integer Programming formulation is given by the following set of equations. We assume that all $\mathbf{l_i}$ are negative and $\mathbf{u_i}$ are positive. In case this isn't true, it is possible to just update the bounds such that they are.

$$
\begin{align}
& \mathbf{l_0} \leq \mathbf{x_0} \leq \mathbf{u_0} \tag{4a}\\
& \hat{\mathbf{x}}_{\mathbf{i+1}} = W_{i+i}\mathbf{x_i} + \mathbf{b_{i+i}} && \forall i \in \{0,\, n-1\} \tag{4b}\\
& \mathbf{x_i} \geq 0 && \forall i \in \{1,\, n-1\} \tag{4c}\\
& \mathbf{x_i} \geq \hat{\mathbf{x}}_{\mathbf{i}} && \forall i \in \{1,\, n-1\} \tag{4d}\\
& \mathbf{x_i} \leq \hat{\mathbf{x}}_{\mathbf{i}} - \mathbf{l_i} \cdot (1 - \boldsymbol{\delta_i}) && \forall i \in \{1,\, n-1\} \tag{4e}\\
& \mathbf{x_i} \leq \mathbf{u_i} \cdot \boldsymbol{\delta_i} && \forall i \in \{1,\, n-1\} \tag{4f}\\
& \boldsymbol{\delta_i} \in \{0,1\}^{h_i} && \forall i \in \{1,\, n-1\} \tag{4g}\\
& \hat{x}_n \leq 0 \tag{4h}
\end{align}
$$

The level 0 of the Sherali-Adams hierarchy of relaxation Sherali & Adams [6] doesn't include any additional constraints. Indeed, polynomials of degree 0 are simply constants and their multiplication with existing constraints followed by linearization therefore doesn't add any new constraints. As a result, the feasible domain given by the level 0 of the relaxation corresponds simply to the removal of the integrality constraints:

$$
\begin{align}
& \mathbf{l_0} \leq \mathbf{x_0} \leq \mathbf{u_0} \tag{5a}\\
& \hat{\mathbf{x}}_{\mathbf{i+1}} = W_{i+i}\mathbf{x_i} + \mathbf{b_{i+i}} && \forall i \in \{0,\, n-1\} \tag{5b}\\
& \mathbf{x_i} \geq 0 && \forall i \in \{1,\, n-1\} \tag{5c}\\
& \mathbf{x_i} \geq \hat{\mathbf{x}}_{\mathbf{i}} && \forall i \in \{1,\, n-1\} \tag{5d}\\
& \mathbf{x_i} \leq \hat{\mathbf{x}}_{\mathbf{i}} - \mathbf{l_i} \cdot (1 - \boldsymbol{d_i}) && \forall i \in \{1,\, n-1\} \tag{5e}\\
& \mathbf{x_i} \leq \mathbf{u_i} \cdot \boldsymbol{d_i} && \forall i \in \{1,\, n-1\} \tag{5f}\\
& \underline{0 \leq \boldsymbol{d_i} \leq 1} && \forall i \in \{1,\, n-1\} \tag{5g}\\
& \hat{x}_n \leq 0 \tag{5h}
\end{align}
$$

Combining the equations (5e) and (5f), looking at a single unit $j$ in layer $i$, we obtain:

Figure 3: *Feasible domain corresponding to the Planet relaxation for a single ReLU.*

$$x_{i[j]} \leq \min \left( \hat{x}_{i[j]} - l_i(1 - d_{i[j]}), u_{i[j]} d_{i[j]} \right) \tag{6}$$

The function mapping $d_{i[j]}$ to an upperbound of $x_{i[j]}$ is a minimum of linear functions, which means that it is a concave function. As one of them is increasing and the other is decreasing, the maximum will be reached when they are both equals.

$$\hat{x}_{i[j]} - l_{i[j]}(1 - d^{\star}_{i[j]}) = u_{i[j]} d^{\star}_{i[j]}$$
$$\Leftrightarrow \qquad d^{\star}_{i[j]} = \frac{\hat{x}_{i[j]} - l_{i[j]}}{u_{i[j]} - l_{i[j]}} \tag{7}$$

Plugging this equation for $d^{\star}$ into Equation(6) gives that:

$$x_{i[j]} \leq u_{i[j]} \frac{\hat{x}_{i[j]} - l_{i[j]}}{u_{i[j]} - l_{i[j]}} \tag{8}$$

which corresponds to the upper bound of $x_{i[j]}$ introduced for Planet [2].

## 4  MaxPooling

For space reason, we only described the case of ReLU activation function in the main paper. We now present how to handle MaxPooling activation, either by converting them to the already handled case of ReLU activations or by introducing an explicit encoding of them when appropriate.

### 4.1  Mixed Integer Programming

Similarly to the encoding of ReLU constraints using binary variables and bounds on the inputs, it is possible to similarly encode MaxPooling constraints. The constraint

$$y = \max \left( x_1, \ldots, x_k \right) \tag{9}$$

can be replaced by

$$y \geq x_i \qquad\qquad \forall i \in \{1 \ldots k\} \tag{10a}$$
$$y \leq x_i + (u_{x_{1:k}} - l_{x_i})(1 - \delta_i) \qquad\qquad \forall i \in \{1 \ldots k\} \tag{10b}$$
$$\sum_{i \in \{1 \ldots k\}} \delta_i = 1 \tag{10c}$$
$$\delta_i \in \{0, 1\} \qquad\qquad \forall i \in \{1 \ldots k\} \tag{10d}$$

where $u_{x_{1:k}}$ is an upper-bound on all $x_i$ for $i \in \{1 \ldots k\}$ and $l_{x_i}$ is a lower bound on $x_i$.

### 4.2  Reluplex

In the version introduced by [4], there is no support for MaxPooling units. As the canonical representation we evaluate needs them, we provide a way of encoding a MaxPooling unit as a combination of Linear function and ReLUs.

To do so, we decompose the element-wise maximum into a series of pairwise maximum

$$\max \left( x_j, x_2, x_3, x_4 \right) = \max( \ \max \left( x_1, x_2 \right),$$
$$\max \left( x_3, x_4 \right)) \tag{11}$$

and decompose the pairwise maximums as sum of ReLUs:

$$\max\left(x_1, x_2\right) = \max\left(x_1 - x_2,\ 0\right) + \max\left(x_2 - l_{x_2}, 0\right) + l_{x_2}, \tag{12}$$

where $l_{x_2}$ is a pre-computed lower bound of the value that $x_2$ can take.

As a result, we have seen that an elementwise maximum such as a MaxPooling unit can be decomposed as a series of pairwise maximum, which can themselves be decomposed into a sum of ReLUs units. The only requirement is to be able to compute a lower bound on the input to the ReLU, for which the methods discussed in the paper can help.

### 4.3 Planet

As opposed to Reluplex, Planet Ehlers [2] directly supports MaxPooling units. The SMT solver driving the search can split either on ReLUs, by considering separately the case of the ReLU being passing or blocking. It also has the possibility on splitting on MaxPooling units, by treating separately each possible choice of units being the largest one.

For the computation of lower bounds, the constraint

$$y = \max\left(x_1, x_2, x_3, x_4\right) \tag{13}$$

is replaced by the set of constraints:

$$y \geq \mathbf{x}_i \qquad \forall i \in \{1 \ldots 4\} \tag{14a}$$

$$y \leq \sum_i \left(x_i - l_{x_i}\right) + \max_i l_{x_i}, \tag{14b}$$

where $x_i$ are the inputs to the MaxPooling unit and $l_{x_i}$ their lower bounds.

## 5 Mixed Integers Variants

### 5.1 Encoding

Several variants of encoding are available to use Mixed Integer Programming as a solver for Neural Network Verification. As a reminder, in the main paper we used the formulation of Tjeng & Tedrake [7]:

$$x_i = \max\left(\hat{\mathbf{x}}_\mathbf{i}, 0\right) \quad \Rightarrow \quad \boldsymbol{\delta_i} \in \{0, 1\}^{h_i}, \quad \mathbf{x_i} \geq 0, \qquad \mathbf{x_i} \leq \mathbf{u_i} \cdot \boldsymbol{\delta_i} \tag{15a}$$

$$\mathbf{x_i} \geq \hat{\mathbf{x}}_\mathbf{i}, \qquad \mathbf{x_i} \leq \hat{\mathbf{x}}_\mathbf{i} - \mathbf{l_i} \cdot \left(1 - \boldsymbol{\delta_i}\right) \tag{15b}$$

An alternative formulation is the one of Lomuscio & Maganti [5] and Cheng et al. [1]:

$$x_i = \max\left(\hat{\mathbf{x}}_\mathbf{i}, 0\right) \quad \Rightarrow \quad \boldsymbol{\delta_i} \in \{0, 1\}^{h_i}, \quad \mathbf{x_i} \geq 0, \qquad \mathbf{x_i} \leq \mathbf{M_i} \cdot \boldsymbol{\delta_i} \tag{16a}$$

$$\mathbf{x_i} \geq \hat{\mathbf{x}}_\mathbf{i}, \qquad \mathbf{x_i} \leq \hat{\mathbf{x}}_\mathbf{i} - \mathbf{M_i} \cdot \left(1 - \boldsymbol{\delta_i}\right) \tag{16b}$$

where $\mathbf{M_i} = \max\left(-\mathbf{l_i}, \mathbf{u_i}\right)$. This is fundamentally the same encoding but with a sligthly worse bounds that is used, as one of the side of the bounds isn't as tight as it could be.

### 5.2 Obtaining bounds

The formulation described in Equations (15) and (16) are dependant on obtaining lower and upper bounds for the value of the activation of the network.

**Interval Analysis** One way to obtain them, mentionned in the paper, is the use of interval arithmetic [3]. If we have bounds $\mathbf{l_i}, \mathbf{u_i}$ for a vector $\mathbf{x_i}$, we can derive the bounds $\hat{\mathbf{l}}_{\mathbf{i+1}}, \hat{\mathbf{u}}_{\mathbf{i+1}}$ for a vector $\hat{\mathbf{x}}_{\mathbf{i+1}} = W_{i+1}\mathbf{x_i} + b_{i+1}$

$$\hat{l}_{i+1[j]} = \sum_k \left( W^+_{i+1[j,k]} l^+_{i[k]} + W^-_{i+1[j,k]} u^+_{i[k]} \right) + b_{i+1[j]} \tag{17a}$$

$$\hat{u}_{i+1[j]} = \sum_k \left( W^+_{i+1[j,k]} u^+_{i[k]} + W^-_{i+1[j,k]} l^+_{i[k]} \right) + b_{i+1[j]} \tag{17b}$$

with the notation $a^+ = \max(a, 0)$ and $a^- = \min(a, 0)$. Propagating the bounds through a ReLU activation is simply equivalent to applying the ReLU to the bounds.

**Planet Linear approximation**  An alternative way to obtain bounds is to use the relaxation of Planet. This is the methods that was employed in the paper: we build incrementally the network approximation, layer by layer. To obtain the bounds over an activation, we optimize its value subject to the constraints of the relaxation.

Given that this is a convex problem, we will achieve the optimum. Given that it is a relaxation, the optimum will be a valid bound for the activation (given that the feasible domain of the relaxation includes the feasible domains subject to the original constraints).

Once this value is obtained, we can use it to build the relaxation for the following layers. We can build the linear approximation for the whole network and extract the bounds for each activation to use in the encoding of the MIP. While obtaining the bounds in this manner is more expensive than simply doing interval analysis, the obtained bounds are better.

## 5.3 Objective function

In the paper, we have formalised the verification problem as a satisfiability problem, equating the existence of a counterexample with the feasibility of the output of a (potentially modified) network being negative.

In practice, it is beneficial to not simply formulate it as a feasibility problem but as an optimization problem where the output of the network is explicitly minimized.

## 5.4 Comparison

We present here a comparison on CollisionDetection and ACAS of the different variants.

1. **Planet-feasible** uses the encoding of Equation (15), with bounds obtained based on the planet relaxation, and solve the problem simply as a satisfiability problem.

2. **Interval** is the same as **Planet-feasible**, except that the bounds used are obtained by interval analysis rather than with the Planet relaxation.

3. **Planet-symfeasible** is the same as **Planet-feasible**, except that the encoding is the one of Equation (16).

4. **Planet-opt** is the same as **Planet-feasible**, except that the problem is solved as an optimization problem. The MIP solver attempt to find the global minimum of the output of the network. Using Gurobi's callback, if a feasible solution is found with a negative value, the optimization is interrupted and the current solution is returned. This corresponds to the version that is reported in the main paper.

The comparison also include two variants of **BlackBox**: **BlackBox** and **BlackBoxNoOpt**. Similarly to **Planet-opt**, **BlackBox** attempts to do global optimization and interrupt the search when a feasible solution with negative value is found. **BlackBoxNoOpt** works like the other MIP encoding by simply encoding the problem as satisfiability.

The first observation that can be made is that when we look at the CollisionDetection dataset in Figure 4a, only **Planet-opt** and **BlackBox** solves the dataset to 100% accuracy. The reason why the other methods don't reach it is not because of timeout but because they return spurious counterexamples. As they encode only satisfiability problem, they terminate as soon as they identify a solution with a value of zero. Due to the large constants involved in the big-M, those solutions are sometimes not actually valid counterexamples. This is a significant advantage to encoding the problem as optimization problems versus simply as satisfiability problems.

The other results that we can observe is the impact of the quality of the bounds when the networks get deeper, and the problem becomes therefore more complex, such as in the ACAS dataset. **Interval** has the worst bounds and is much slower than the other methods. **Planetsym-feasible**, with its slightly worse bounds, performs worse than **Planet-feasible** and **Planet-opt**.

|                            |                         |
| :------------------------: | :---------------------: |
| (a) CollisionDetection Dataset | (b) ACAS Dataset |

Figure 4: *Comparison between the different variants of MIP formulation for Neural Network verification.*

# 6 Experimental setup details

We provide in Table 1 the characteristics of all of the datasets used for the experimental comparison in the main paper.

| Data set | Count | Model Architecture |
| :------: | :---: | :----------------: |
| Collision Detection | 500 | 6 inputs<br>40 hidden unit layer, MaxPool<br>19 hidden unit layer<br>2 outputs |
| ACAS | 188 | 5 inputs<br>6 layers of 50 hidden units<br>5 outputs |
| PCAMNIST | 27 | 10 or {5, 10, 25, 100, 500, 784} inputs<br>4 or {2, 3, 4, 5, 6, 7} layers<br>of 25 or {10, 15, 25, 50, 100} hidden units,<br>1 output, with a margin of +1000 or<br>{-1e4, -1000, -100, -50, -10, -1 ,1, 10, 50, 100, 1000, 1e4} |

Table 1: *Dimensions of all the data sets. For PCAMNIST, we use a base network with 10 inputs, 4 layers of 25 hidden units and a margin of 1000. We generate new problems by changing one parameter at a time, using the values inside the brackets.*

# 7 Additional performance details

Given that there is a significant difference in the way verification works for SAT problems vs. UNSAT problems, we report also comparison results on the subset of data sorted by decision type.

# 8 PCAMNIST details

**PCAMNIST** is a novel data set that we introduce to get a better understanding of what factors influence the performance of various methods. It is generated in a way to give control over different architecture parameters. The networks takes $k$ features as input, corresponding to the first $k$ eigenvectors of a Principal Component Analysis decomposition of the digits from the MNIST data set. We also vary the depth (number of layers), width (number of hidden unit in each layer) of the networks. We train a different network for each combination of parameters on the task of predicting the parity of the presented digit. This results in the accuracies reported in Table 2.

(a) *On SAT properties*          (b) *On UNSAT properties*

Figure 5: *Proportion of properties verifiable for varying time budgets depending on the methods employed on the CollisionDetection dataset. We can identify that all the errors that **BlackBox** makes are on SAT properties, as it returns incorrect counterexamples.*

(a) *On SAT properties*          (b) *On UNSAT properties*

Figure 6: *Proportion of properties verifiable for varying time budgets depending on the methods employed on the ACAS dataset. We observe that **planet** doesn't succeed in solving any of the SAT properties, while our proposed methods are extremely efficient at it, even if there remains some properties that they can't solve.*

The properties that we attempt to verify are whether there exists an input for which the score assigned to the odd class is greater than the score of the even class plus a large confidence. By tweaking the value of the confidence in the properties, we can make the property either True or False, and we can choose by how much is it true. This gives us the possibility of tweaking the "margin", which represent a good measure of difficulty of a network.

In addition to the impact of each factors separately as was shown in the main paper, we can also look at it as a generic dataset and plot the cactus plots like for the other datasets. This can be found in Figure 7

Figure 7: *Proportion of properties verifiable for varying time budgets depending on the methods employed. The PCAMNIST dataset is challenging as None of the methods reaches more than 50% success rate.*

| Network Parameter | | | Accuracy | |
|---|---|---|---|---|
| Nb inputs | Width | Depth | Train | Test |
| 5 | 25 | 4 | 88.18% | 87.3% |
| 10 | 25 | 4 | 97.42% | 96.09% |
| 25 | 25 | 4 | 99.87% | 98.69% |
| 100 | 25 | 4 | 100% | 98.77% |
| 500 | 25 | 4 | 100% | 98.84% |
| 784 | 25 | 4 | 100% | 98.64% |
| 10 | 10 | 4 | 96.34% | 95.75% |
| 10 | 15 | 4 | 96.31% | 95.81% |
| 10 | 25 | 4 | 97.42% | 96.09% |
| 10 | 50 | 4 | 97.35% | 96.0% |
| 10 | 100 | 4 | 97.72% | 95.75% |
| 10 | 25 | 2 | 96.45% | 95.71% |
| 10 | 25 | 3 | 96.98% | 96.05% |
| 10 | 25 | 4 | 97.42% | 96.09% |
| 10 | 25 | 5 | 96.78% | 95.9% |
| 10 | 25 | 6 | 95.48% | 95.2% |
| 10 | 25 | 7 | 96.81% | 96.07% |

Table 2: Accuracies of the network trained for the PCAMNIST dataset.