[Reviews · NeurIPS 2018]

Reviewer 1



This paper presents a branch-and-bound based unified framework for piecewise linear neural network (PLNN) verification, and demonstrates how two existing methods (Reluplex and Planet) can be cast in these common terms. It explores other variations of various components of B-and-B, finding that a "smart branching (SB)" heuristic inspired by Kotler and Wong [12] results in a substantially faster method. My take on this paper is somewhat lukewarm, given the mix of strengths and weaknesses it has. STRENGTHS: The paper is very well-written, at least the first half of it. It motivates the problem well, sets the scope of the work (what it covers, what it doesn't), and summarizes the contributions in a meaningful way. The background text is easy for follow, especially for anyone with some background in optimization, MIP, branch-and-bound, etc. It's a dense paper (which is also a weakness), with many details of the encodings described in the paper and the supplementary material. Being able to see existing methods (well, two existing methods) in the light of a MIP-based branch-and-bound framework is useful in understanding how they relate and which other decision points might be worth exploring. The experiments reveal that the proposed BaB-SB method clearly outperforms various baselines in the ACAS dataset. WEAKNESSNES: It's very confusing (somewhat disturbing) to me that you found Gurobi, a commercial MIP solver, to produce "invalid" counter-examples in about 12% of the CollisionDetection cases (going by Fig 2(a)) when Gurobi is running, in a sense, on the purest setting of the problem, Eq (2). Numerical instabilities are well-known in the discrete optimization literature, but over 10% error rate is unheard of. This makes me wonder about two things: * In section 3.1, the description is imprecise as to whether the case of global minimum being zero is a proper counter-example or not. That is, do you mean 'negative' (less than 0) or 'non-positive' (less than or equal to 0) when encoding a counter example? Note that MIP solvers cannot encode strict inequalities of the form "a strictly less than b". * If the error cases are when the generated (but invalid) counter-example evaluates exactly to 0, and that's errorneous because of numerical instability, would it be possible for you to try adding a little negative epsilon (e.g., -0.001) and searching for examples evaluate no more than that? It may very well be that Gurobi on Eq (2) is slow, but if encoded correctly, it should not be inaccurate. I wonder if this affects any other findings too. The extent of technical novelty is somewhat limited. The b-and-b framework of Algorithm 1 is, of course, very standard in discrete optimization literature, even if new to some NIPS readers; what's new here is its application to the verification problem formulated in Eq (3). In terms of techniques, the BaB-SB system is interesting, although it wasn't immediately clearly how difficult was it to adapt the work of Kotler and Wong [12] to the smart branching heuristic used here. One aspect that was confusing for me is that Reluplex is described as maintaining a complete assignment to all variables at all times, and flipping values to try to satisfy more and more constraints. This is the nature of LOCAL SEARCH solvers, which are a distinct class of solvers in the discrete optimization community than branch-and-bound. It is not immediately clear how natural it is to fit a local search solver into a b-and-b one. The motivation for introducing a new dataset, PCAMNIST, for the analysis in Fig 4 wasn't made clear. Could you have used either of the existing datasets for this purpose? The figures, in general, were very difficult to follow. Larger fonts and larger size (at the expense of less text/caption) would help. It might even be worth cutting a figure or two to use the room to better explain what you do retain. ADDED AFTER AUTHOR RESPONSE: Thank you for the clarifications. It was helpful to follow up on some of them in your supplementary material. Although the space is always tight, it would be valuable if you are able to incorporate explanations for some of these points in the main paper (at the expense of some other parts, of course).

Reviewer 2



The paper studies the problem of formal verification of piecewise linear Neural Network models (PLNNs). The authors (i) nicely reframe the verification problem as a case of Branch-and-bound optimization in order to compare the existing methodologies from the literature, (ii) introduce a Branch and Bound algorithm for verifying properties of piecewise linear neural networks, and (iii) propose algorithmic improvements to speed-up runtime efficiency of their algorithm that are verified empirically. There are two key points I would like to raise: (1) The verification problems that are investigated in the paper are constraint satisfaction problems (with originally no objectives). While it is sufficient to reformulate the CSPs studied in the paper as optimization problems, more complex problems that involve using PLNNs for optimization are ignored and would not work directly in the framework proposed in Section 3.1. For example, planning/control optimization in deep-net learned models [1] would seem to be precluded by Section 3.1. It would be much more general to have a unified formalism that handled PLNN property satisfaction as well as full PLNN optimization, so that branch and bound optimizations would benefit both problems. (2) An advantage of the branch and bound framework is that there are already numerous methods for both branching and bounding. While I can see that bounding methods should probably be specialized for the specific deep networks you are studying, it is less clear to me why the authors do not study a wider range of branching methods from the literature. Further if the authors used a modifiable (and very fast) C implementation of branch and bound provided in the SCIP solver (http://scip.zib.de/), they would be able to simply use a wide variety of pre-implemented (and tuned/debugged) heuristics. So while I feel like the authors have provided an interesting unifying framework, they have fallen short of a comprehensive evaluation of existing and new heuristics that would be highly informative for those working in this area. Being comprehensive it would probably have to go in a supplementary appendix, but I see no problem with this… a summary can be placed in the main paper. Overall, I like this paper and the code release is a *huge* bonus, but I would like it even more if the experimental evaluation were much more comprehensive of existing branching (and maybe other bounding) methods. Reference: [1] Buser Say, Ga Wu, Yu Qing Zhou, Scott Sanner. Proceedings of the Twenty-Sixth International Joint Conference on Artificial Intelligence (IJCAI-2017), 750-756. Note: many of the arxiv references in the paper should be replaced with the actual peer-reviewed publication citations.

Reviewer 3



The main contribution of this paper is to provide a framework for NN verification based on Branch and Bound optimization which allows for reframing existing state-of-the-art verification methods as special cases of this framework. With this unified framework, the authors were able to identify algorithmic improvements in the verification process that lead to a speed-up of almost two orders of magnitude. In order to design a unified framework, the authors convert the verification problem into a global optimization problem. The idea is to convert any boolean formula over linear inequalities into a sequence of linear and max-pooling layers. The verification problem is then reduced to finding out the sign of the global minimum of this network. To find this global minimum, the authors present an approach based on Branch and Bound paradigm. They also show how verification methods based on Satisfiability Modulo Theory (SMT) fit in this framework. A generic branch and bound based algorithm is presented to find a global minimum for the verification problem. The authors also show how some of the published works for NN verification based on SMT theory (like Reluplex by Katz et.al and Planet by Ehlers et.al) can be converted to branch and bound framework for verification. With this unified framework, the authors were able to come up with many promising suggestions to improve the speed-up of verification algorithms which they include in their proposed algorithm. They evaluate their method against SMT based methods like Reluplex and Planet, MIP solvers based like MIPplanet. They use Collision Detection, Airborne Collision Avoidance System (ACAS) and PCAMNIST as their datasets for evaluation. PCAMNIST is a novel dataset introduced by the authors to understand the factors that influence various verification methods. The authors observe that on the CollisionDectection dataset most of the solvers succeeded within 10s. In particular, they found SMT based solvers to be extremely fast on this dataset. On ACAS, most of the solvers could not verify within the timelimit of 2 hours. However, their method was able to achieve 98.40% within an hour as compared to the baseline Reluplex which achieved only 79.26% success within the two hour limit. In addition to the evaluation, the authors do an ablation study on what changes made the most impact in terms of performance on the methods that they proposed. They come up with the following conclusions on their study: a) Smart branching contributes significantly to the performance gap b) Branching over inputs rather than over activations does not contribute much improvement c) Better bounds contribute to the rest of the performance gap In their experiments, they also observe the following: a) Larger the number of inputs, harder is the verification problem b) Wider networks take more time to solve as they have more non-linearities c) Properties with large satisfiability margin are easy to prove Quality: This paper proposes a novel framework for NN verification. They make a strong case on how this framework can be instrumental in bringing about rapid improvements in the performance of NN verifications. They provide thorough analysis on what makes their proposed algorithm perform better than the other state-of-the-art approaches. The amount of speed-up that their approach achieves is also significant. Though this approach shows promising results on smaller networks, it would be interesting to see how it scales on larger networks. Overall, the quality of this paper is good. Clarity: The paper is quite clear in presenting the contributions that it makes in the field of verification methods for neural networks. Their main idea has been illustrated with simple examples that are easy to understand. The results and conclusions have been presented well with graphs. However, in Section 5.1 Figure1(b), the y axis is not visible clearly. Originality: The field of formal verification of Neural networks has emerged fairly recently during the last few years. Most of the published works so far have used either SMT theory or MIP formulation to solve the verification problem. To the best of my knowledge, there has been no study that proposes a unified framework for this problem. The authors present a unique and interesting approach that enables us to convert any SMT based verification method or MIP based verification method to a branch and bound based paradigm. This framework makes it easier to come up with solutions to improve the verification process as the authors demonstrate through the various experiments. Significance: Deep neural networks are now increasingly becoming a major component of many safety-critical applications. Thus, formal verification of neural networks has become an important research problem that needs to be solved in order to scale deep neural networks in safety-critical applications. By proposing a unified framework for NN verification, this paper comes up with significant improvements in runtime for verification algorithms. It would be interesting for the research community to analyze this framework which might lead to even more efficient algorithms. A few detailed comments: In Section 5.1 Figure1(b), the y-axis is not visible clearly. In Figure 4. (c) The values on x-axis are not clearly visible. Though this approach shows promising results on smaller networks, it would be good to include larger networks and observe how the algorithm scales. Although the authors in this work focus on piecewise-linear NN’s, adding a few lines as to the feasibility of scaling this framework to general Neural networks might be helpful in motivating other research groups to study this framework further.